# VOC Detections by p-Type Semiconducting Sensors Using Nano-Sized SmFeO_3_ Particles

**DOI:** 10.3390/s22155616

**Published:** 2022-07-27

**Authors:** Masami Mori, Ayumu Noguchi, Yoshiteru Itagaki

**Affiliations:** Graduate School of Science and Engineering, Ehime University, Matsuyama 790-8577, Japan; mori.masami.mm@ehime-u.ac.jp (M.M.); i843021k@mails.cc.ehime-u.ac.jp (A.N.)

**Keywords:** p-type semiconducting sensor, SmFeO_3_, volatile organic compounds (VOCs), ozone

## Abstract

Nano-sized SmFeO_3_ particles are prepared by the pyrolysis of heteronuclear cyano-complex, Sm[Fe(CN)_6_]·4H_2_O at a temperature of 600 °C in ozone. The low temperature decomposition followed in ozone successfully yielded fine particles with a high specific surface area of 20.0 m^3^/g (sample A). The fine particles were classified into further smaller particles with a unimodal size distribution and this process yielded a high specific surface area of 26.0 m^3^/g (sample B). These semiconducting powders were deposited on a sensor electrode by electrophoretic deposition (EPD) and tested on their sensing properties to VOCs. The sensors consisting of samples A and B both showed good responses to ethanol at 285 and 320 °C. The sensor with sample B showed extraordinarily good selectivity of ethanol for toluene at 320 °C. This could be because the detection film of sample B with moderately grown particles selectively reduced the reaction activity of toluene. The sensor with sample B also exhibited good selectivity of ethanol for hexane and dichloromethane.

## 1. Introduction

Environmental monitoring is an urgent task to control ambient air quality at a variety of locations. Analytical techniques are currently adopted for monitoring systems, but their usage is rather restricted due to their high cost and the large size of the equipment. Use of semiconductor gas sensors instead of analytical has aimed at solving these problems. Since integration of sensor elements is inevitable to add a gas recognition function in ambient air, each element is required to be of small size, low cost, and low energy consumption, as well as high sensing performance. Semiconducting gas sensors convert a reaction of a detected gas with the adsorbed oxygen on the surface into a change in electrical resistance. The semiconducting gas sensors have a simple structure and detect a variety of gasses sensitively [1,2]. So far, n-type semiconducting oxides such as SnO_2_ have been widely studied and have already been put into practical use [3]. In the atmosphere, the surface of SnO_2_ particles covered with negatively charged adsorbed oxygen forms the depletion layer near the particle surface, inducing a potential barrier for electron transfer at the grain boundaries. The potential barrier is reduced when the target gas reacts to eliminate the adsorbed oxygen. In the case of p-type semiconducting oxides, negatively charged adsorbed oxygen forms a hole accumulation layer near the particle surface, and holes only migrate through the layer. Therefore, it is generally accepted that p-type semiconducting gas sensors tend to be inferior in sensitivity compared to n-type; it is generally accepted that the sensitivity of a p-type semiconductor is the square root of that of n-type [4]. Some p-type semiconductor oxides, however, have a high oxidation catalytic activity [5,6]. Catalytic reactions taking place on a p-type semiconducting oxide effectively change the hole concentration in the accumulation layer. In fact, previous studies have revealed that p-type perovskite oxides such as LaFeO_3_ [7,8,9,10] and SmFeO_3_ [8,9,10,11,12,13,14] were promising materials to detect pollutant gasses such as NO_x_ with high sensitivity and reliability. A powder of SmFeO_3_ prepared by the thermal decomposition of the cyano-complex, Sm[Fe(CN)_6_]·4H_2_O, has a high surface concentration of Sm^3+^ ions at the A site [9]. The SmFeO_3_ thick films screen-printed on a comb-type electrode showed a high response, specifically to some target gases [15,16,17].

A conventional solid-state synthesis requires temperature of over 1000 °C to obtain a single phase of SmFeO_3_ [12]. As a result, the specific surface area of the SmFeO_3_ particles decreases owing to grain growth during the high temperature decomposition. The thermal decomposition of the cyano-complex in air allows reduction of pyrolysis temperature to 600 °C, and the specific surface area of SmFeO_3_ powder prepared by this method is around 10 m^2^/g [12]. We previously examined sensor characteristics of SmFeO_3_ for volatile organic compounds (VOCs) [15]. The SmFeO_3_ sensor, in that study, exhibited a good response especially to alcohols; *S* = *R*_VOC_/*R*_air_ was around 30. Since VOC sensing characteristics of p-type semiconductors significantly depend on the oxidation activity for VOC, surface area of the oxide powder will be deeply related to the sensor response. To further increase the surface area of SmFeO_3_ powder, we, in this study, used ozone-air mixed gas instead of pure air as an oxidizer of the cyano-complex, since ozone has a much larger oxidizing power than air and is assumed to decompose the cyano-complex rapidly at low temperature. In this study, a SmFeO_3_ powder obtained by the ozone oxidation at 600 °C was found to have a higher surface area than those by conventional air oxidation. We thus examined VOC sensing characteristics of the SmFeO_3_ powders prepared by ozone pyrolysis.

## 2. Materials and Methods

The heteronuclear complex, Sm[Fe(CN)_6_]·4H_2_O, was synthesized at room temperature by mixing aqueous solutions of equimolar amounts of Sm(NO_3_)_3_ and K_3_[Fe(CN)_6_] under continuous stirring. The resulting precipitate was washed with water, ethanol, and diethyl ether, before drying in air at 50 °C. The complex was set in a tubular furnace and heated at 600 °C for 2 h under ozone gas flow at a rate of 100 mL/min. Ozone gas was generated from compressed air with an ozone gas generator (Funatech SO-250). The products were measured by X-ray diffraction (XRD; X’pert Pro MPD using a Cu Kα radiation, PANalytical) and confirmed the formation of a single phase of the SmFeO_3_. The obtained SmFeO_3_ powder was crushed by wet ball milling in ethanol and dried in an oven (sample A). The powder of sample A was then dispersed in acetyl acetone and sonicated for 1 h at room temperature. The suspension was left to stand for 1h to separate it into the components of supernatant and precipitate. The particles in the supernatant layer were collected by a centrifugal separation (sample B). The surface area of the powders was measured by BET analysis using N_2_ adsorbent (fine precision surface area and porosimetry system, BELsorp-mini, microtracBEL) and the particle size was analyzed by FE-SEM (S-5500, HITACHI).

Alumina sheets (1.0 mm × 1.0 mm) attached with Pt counter electrodes and microheater were used as a sensor substrate. SmFeO_3_ powder was deposited on the Pt electrode by the EPD method. To prepare an organic suspension of the oxide particle for EPD, 0.04 g of SmFeO_3_ was dispersed in 10 mL of acetylacetone and 2.5 mg of I_2_ was added as a charging agent. The added I_2_ reacts with the solvents to form HI and the free protons generated by HI dissociation are adsorbed on the suspended SmFeO_3_ particle [18]. The mixture was stirred and then sonicated for 30 min at room temperature to obtain a good dispersion. The positively charged SmFeO_3_ particle is expected to migrate and deposit on the cathodic electrode during the EPD process. The deposition was performed at constant DC voltage of 3 or 5 V for 1–3 min. After the EPD process, the deposits were dried at 70 °C in a drying oven for one day, and then sintered at 600 °C for 3 h with an electric furnace. The surface of the sintered films was observed by FE-SEM.

Figure 1 shows the setup for the sensor tests. The sensor was placed in the stainless chamber and the Pt electrodes and Pt microheater were connected to a d.c. power and a digital electrometer as a heater power source and a resistance meter (Advantest Co., TR8652, Tokyo, Japan). Test gasses were evaporated at a constant temperature from a diffusion tube loaded in a permeator (GASTECH, Co., PD-1B, Ayase, Japan). Compressed air from an oil-free air compressor was introduced in the permeator along with a water bubbler to control humidity of the test gasses at 50% RH. Test gasses and air were alternately introduced in the chamber with a flow rate of 400 mL/min. The conductance of the elements was measured using a digital electrometer (Advantest, TR8652) at different gas concentrations and operating temperatures.

## 3. Results

### 3.1. Preparation of SmFeO_3_ Nanoparticles

The XRD pattern of sample A shown in Figure 2 indicates that a single phase of SmFeO_3_ perovskite was formed by the pyrolysis of the complex at 600 °C in ozone.

Figure 3 shows the volume distributions of the sample powders. Sample A possesses two peaks of distributions centered at 0.40 and 5.5 μm, and the smaller component was predominant. The powder of sample B possesses nearly single modal distribution peaked around 0.17 μm. Median diameters of the powders were evaluated as follows: 0.44 μm (sample A) > 0.17 μm (sample B).

Figure 4 shows the nitrogen adsorption isotherms of the powder samples. The adsorption isotherm of sample A shows an intermediate pattern between type II and type IV in which the presence of slight hysteresis can be confirmed, along with macropores (pores of 50 nm or larger) and mesoporous adsorbent (pores of 2 to 50 nm). The hysteresis obtained from sample A is H3 type (IUPAC classification) [19,20], which is a pattern often seen in plate-like aggregates. The powder of sample B can also be judged to be mesoporous adsorbent. The isotherm of this powder shows low pressure hysteresis with no closing point of the hysteresis loop. From the above results, sample A contains a mixture of mesopore particles with different surface areas. The classification operation separated the mixture in sample A into sample B, with a large surface area, and the residue. From the nitrogen adsorption isotherm, specific surface area of samples A and B were evaluated with the BET equation, resulting in 20.0 m^3^/g for sample A and 26.0 m^3^/g for sample B. It should be noted that specific surface area of the SmFeO_3_ powder before milling was 17.9 m^2^/g.

Figure 5 shows FE-SEM photographs of samples A and B. SmFeO_3_ particles with a particle size of 50 nm or smaller were observed for the both samples.

### 3.2. Evaluation of Sensor Response Characteristics

In this study the powders of samples A and B were tested as sensor materials. Figure 6 shows SEM photographs of the surface of the sensing layers sintered at 600 °C. The particle size of the powder was smaller in sample B than in sample A, but when the film was formed on the sensor substrate by the EPD method and fired at 600 °C, particle growth due to aggregation occurred. The surface of sample A shows aggregation of particles with a size of 50–60 nm, whereas the particles of sample B clearly coarsened with the neck growth. The detection layers were prepared by the EPD method three time for each sample, resulting in similar morphologies with those in Figure 6.

Resistance of SmFeO_3_ sensors in air is in the range of 1 × 10^5^–1 × 10^7^ Ω at operating temperatures [15]. The present sensors were in the resistance range of 10^7^ and 10^6^ Ω in air at 285 and 320 °C, respectively. For this reason, we, in this study, set temperature of sensor operation to 285 and 320 °C. Figure 7 shows the transient response curves of the sensors in 5 ppm ethanol at 320 °C. The resistance value of sample A in air was 1 × 10^7^ Ω, and that of sample B in air was 1 × 10^6^ Ω. The difference in the resistance in air may be ascribed to the surface morphology, where sample B showed a more sintered phase. It should be noted that resistance values in dry air are about a half of those in wet air. Li et al. [21] reported on the humidity property of MoS_2_@CuO heterogenous photocatalyst. They proposed that the separated holes by photo-illumination are captured by water to form proton and hydroxyl radical Equation (1). This reaction, if taking place on the surface of SmFeO_3_, would increase resistance of SmFeO_3_. At present, we are investigating the humidity dependence of the VOC sensing property.
h^+^ + H_2_O → H^+^ + OH (1)

The 90% response time was 1–2 min, but the recovery time when switching from ethanol to air was very slow, i.e., more than 20 min to return to the resistance in air. Both the sensors exhibited rapid responses from air to ethanol. Recovery from ethanol to air was very slow and recovery time is >20 min. This is due to reaction products, CO_2_ and/or H_2_O, strongly adsorbed on the surface of the SmFeO_3_ particles. It should be noted that a signal fluctuation was observed in Figure 7 under the air flow. The cause of this phenomenon is still unknown, but possibly due to an instability of the electrometer induced by an interference from peripheral devices such as a switching timer or others.

Figure 8 shows the responses of the sensors with samples A and B in ethanol and toluene at 285 and 320 °C. The sensor responses are higher at 285 °C than at 320 °C in the sensor with sample A, and vice versa in sample B. The response of sample A to 5 ppm ethanol reached 145, and that in 5 ppm toluene was about 60. On the other hand, the sensor response with sample B is about 125 to 5 ppm ethanol at 320 °C, but the response to 5 ppm toluene is no more than 25 which is much lower than that in the sensor with sample A. It was thus revealed that a SmFeO_3_ fine particles (samples A and B) can both detect ppm level ethanol with the outstandingly high response. Sample B is more effective to detect ethanol selectively for toluene. The power law is widely accepted in semiconducting sensor where sensor resistance is related to the power of gas concentration [22], hence *R* = *aP^n^*, where *a* and *n* are constants, and *P* is partial pressure of a target gas. The response curves in Figure 8 were thus re-plotted in Figure 9 with a log-log plot to obtain the constants in the power law. The log-log plot showed a good linearity and constants *a* and *n* were obtained as summarized in Table 1. Clearly *a*-values correlate to sensitivity to each VOC as seen in the concentration dependence curves in Figure 8. It seems that the difference in *a*-values of sample A is smaller than that of sample B. The *a*-value of the sensor with sample B in ethanol at 320 °C is specifically large, resulting in the high selectivity to ethanol at 320 °C. As regards *n*-values, those of sample A are close to 1 except for toluene at 320 °C. If VOCs in gas phase behave according to the Henry’s law, the amount of adsorption of VOCs is expected to be proportional to partial pressure of the gasses. The sensor with sample A shows responses nearly proportional to concentration of VOCs, suggesting that surface adsorption process of VOC or reaction products are predominant in the sensor mechanism. The fact that the sensor response increases with decreasing temperature is consistent with the above mechanism. On the other hand, the *n*-values of sample B are significantly smaller than those of sample A, and the sensor response increases with increasing temperature in the case of sample B. This indicates that oxidation reaction of the VOCs is rather dominant in sample B. Although sample B possesses larger specific surface area than sample A, it moderately coarsened during the heating process at 600 °C. This particle growth may reduce catalytic activity to both VOCs, especially to toluene. We previously examined catalytic activity and sensor response of Sm_2_O_3_ loaded SmFeO_3_ powders to 30 ppm ethanol and toluene [23]. The catalytic activity of toluene and the sensor response largely depended on the amount of Sm_2_O_3_ added.

This indicates that Sm^3+^ acts as an adsorption site of toluene and catalytic activity and sensor response strongly depends on the amount of toluene. On the other hand, ethanol, having slight acidity, may adsorb more favorably on the Sm^3+^ ion, having slight basicity, regardless of the number of Sm^3+^ ions on the surface. The sensing mechanism of VOC on the different morphology of semiconducting film should be discussed more quantitatively. The present research, however, gave us one qualitative conclusion: that a slightly agglomerated surface effectively suppresses the amount of toluene adsorption, but this is not the case in ethanol, and provides a large difference in sensor response between ethanol and toluene. Figure 10 shows the sensor responses of the sensor with sample B to 4 types of VOCs: ethanol (alcohol), hexane (aliphatic hydrocarbon), dichloromethanes (organic halide), and toluene (aromatic hydrocarbon). The sensor with sample B also exhibited high selectivity to ethanol among these VOCs, especially at 320 °C.

### 3.3. VOC Detection Using p-Type Semiconducting Oxides

VOC detection has been conducted using p-type semiconducting oxides such as NiO [24,25], Co_3_O_4_ [26] and some perovskite-type oxides [27]. Carbone et al. [24] reported a sensor response of *R*_VOC_/*R*_air_ = 35 at 200 °C to 150 ppm ethanol using NiO grained flowers having a significantly high specific surface area of 240.3 m^2^/g. Kruefu et al. [25] obtained the response value of around 40 at 350 °C to 2000 ppm ethanol sensing using Ru-loaded NiO particles with a dimension range of 10–40 nm. Zhang et al. [26] reported the response of 221.99 to 100 ppm ethanol using porous Co_3_O_4_ with a specific surface area of 28.509 m^2^/g. For perovskite-type oxides, LaFeO_3_ has been tested by some research groups [27,28]. Dai et al. [27] prepared LaFeO_3_ thin film with the periodic pores of 500 nm in dimension and yielded the response of 14 to 5 ppm ethanol at 450 °C. Yu et al. [28] prepared the Ag-modified LaFeO_3_ with a specific surface area of 16.45 m^2^/g and obtained the response value of 20.9 to 20 ppm ethanol at 180 °C. It is thus obvious that high specific surface area or small particle size of p-type semiconducting oxides is a critical factor for a highly sensitive VOC detection. We previously conducted VOC sensing using the SmFeO_3_ particle [15]. The SmFeO_3_ particles possessed the specific surface area of 21.6 m^2^/g after the pyrolysis of the cyano-complex at 900 °C in air and the ball-mill crushing process. The sensor response to 3 ppm ethanol was 30 at 400 °C. The SmFeO_3_ sensor exhibited an outstandingly high response to ethanol of a low concentration. The SmFeO_3_ powder, in this study, prepared by the ozone-decomposition possesses a comparable surface area without the ball-milling process, that is 17.9 m^2^/g. The SmFeO_3_ sensor based on the ozone-decomposed powder exhibited an even higher response to ethanol, as shown in Figure 10. This suggests that specific surface area is not a sole factor in sensor response. It is predictable that the SmFeO_3_ particles prepared by the ozone decomposition possess a more favorable surface compared to those by air decomposition.

## 4. Conclusions

Nano-sized particles of SmFeO_3_ with larger specific surface area were obtained by pyrolyzing cyano-complex, Sm[Fe(CN)_6_]·4H_2_O, in ozone at a low temperature of 600 °C. The obtained fine particles possessed a bimodal volume distribution (sample A) and were further classified into finer particles with a unimodal one (sample B). By using the EPD method in the organic suspension with the applied voltage of 3–5 V, SmFeO_3_ particles were uniformly deposited on the tiny electrode with a dimension of 1.0 × 1.0 mm. The two sensors with samples A and B were tested to detect ethanol and toluene at 285 and 320 °C. The sensors exhibited high responses to ethanol, at around *R*_VOC_/*R*_air_ = 140 at 5 ppm. However, the sensor sample B exhibited a specifically high selectivity to ethanol at 320 °C. We tentatively speculate that the moderately agglomerated morphology of the sample B film selectively reduced the amount of adsorption of toluene. The sensor also exhibited good selectivity of ethanol to hexane and dichloromethane.

## Figures and Tables

**Figure 1 sensors-22-05616-f001:**
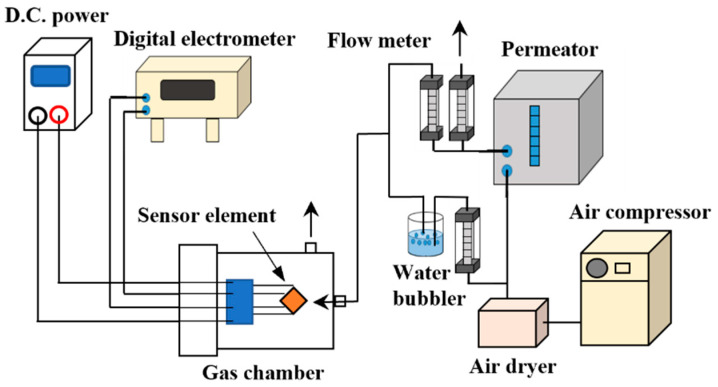
Experimental setup for the sensor test.

**Figure 2 sensors-22-05616-f002:**
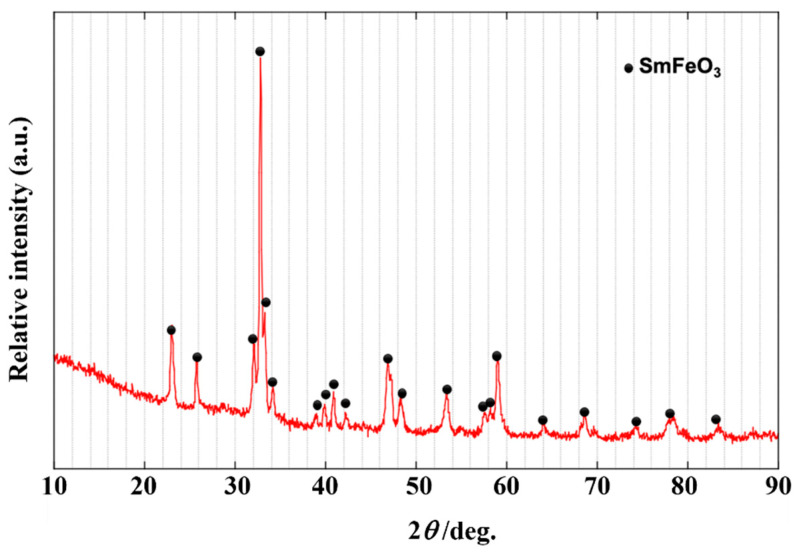
XRD patterns of the SmFeO_3_ powder (sample A).

**Figure 3 sensors-22-05616-f003:**
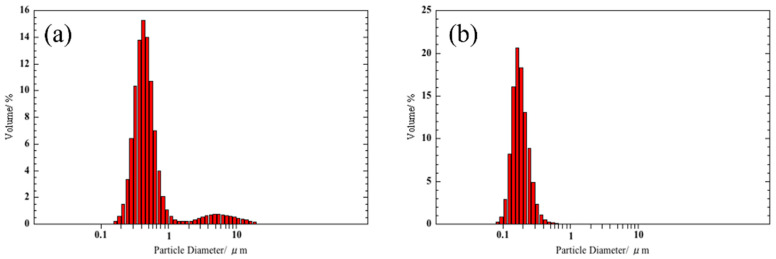
Volume distribution of the classified SmFeO_3_ powders: (**a**) as balled milled (sample A), and (**b**) the supernatant component of the suspension of sample A (sample B).

**Figure 4 sensors-22-05616-f004:**
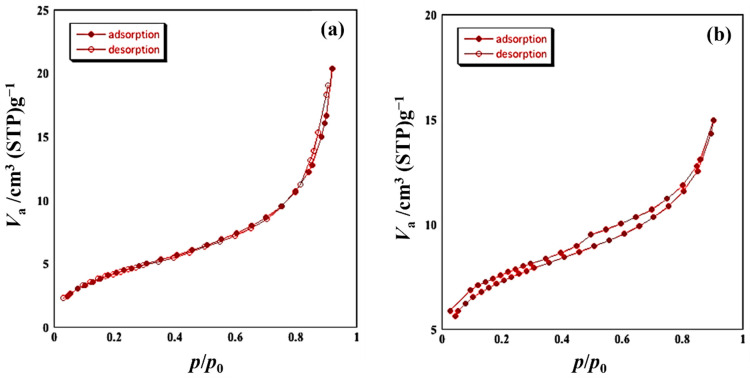
Nitrogen adsorption isotherms of samples A (**a**) and B (**b**) measured at 77K.

**Figure 5 sensors-22-05616-f005:**
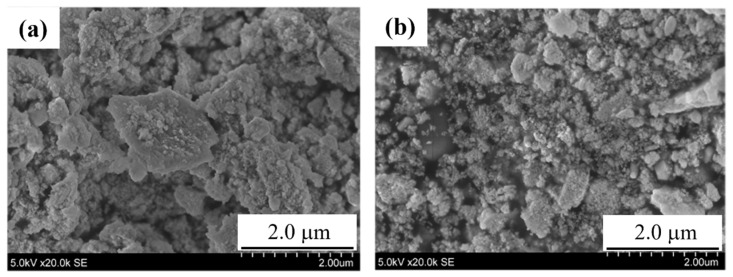
FE-SEM images of the powders of samples A (**a**) and B (**b**).

**Figure 6 sensors-22-05616-f006:**
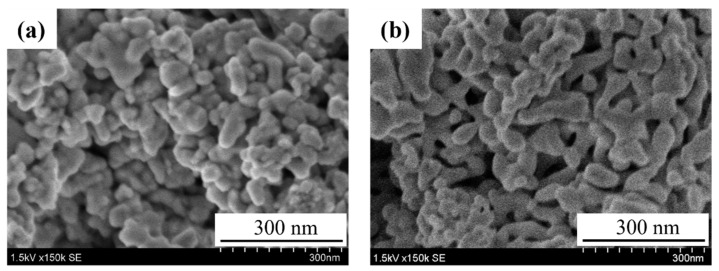
SEM images of the surface of the SmFeO_3_ detection layers of sample A (**a**) and B (**b**) sintered at 600 °C.

**Figure 7 sensors-22-05616-f007:**
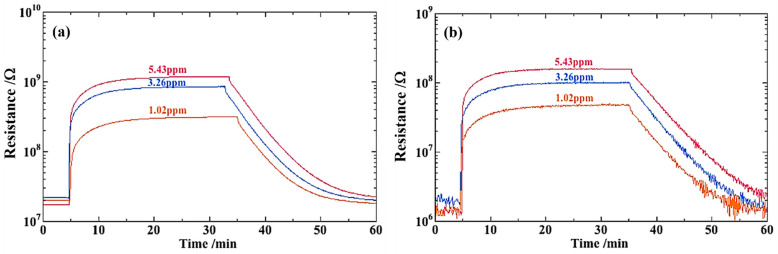
Transient response curves of the sensors with sample A (**a**) and B (**b**) in ethanol at 320 °C.

**Figure 8 sensors-22-05616-f008:**
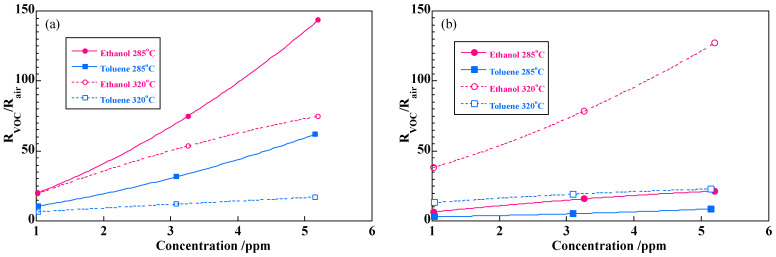
Concentration dependences of sensors responses (*R*_VOC_/*R*_air_) in ethanol and toluene: (**a**) sample A and (**b**) sample B at 285 and 320 °C.

**Figure 9 sensors-22-05616-f009:**
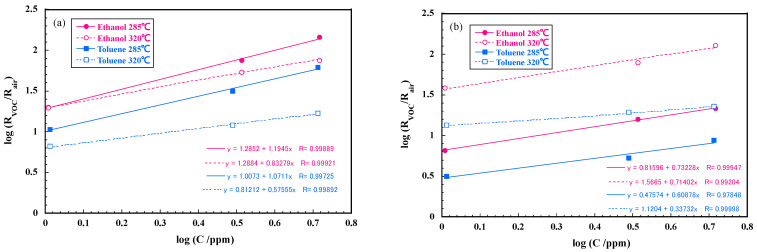
Log-log plots of concentration of ethanol and toluene vs. sensor responses at 285 and 320 °C with sample A (**a**) and sample B (**b**).

**Figure 10 sensors-22-05616-f010:**
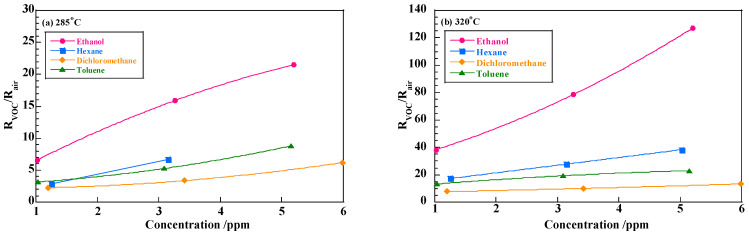
Sensor responses to VOCs with sample B at 285 °C and 320 °C.

**Table 1 sensors-22-05616-t001:** Characteristic parameters of *a* and *n* assuming *R*_VOC_/*R*_air_ = *aP^n^* for the sensors with samples A and B.

Target Gas	Temp. (°C)	Sample A		Sample B	
		*a*	*n*	*a*	*n*
Ethanol	285	19.3	1.19	6.55	0.732
	320	19.4	0.833	36.9	0.714
Toluene	285	10.2	1.07	2.99	0.609
	320	6.49	0.576	13.2	0.337

## Data Availability

Not applicable.

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
