# Peer review of "VOC Detections by p-Type Semiconducting Sensors Using Nano-Sized SmFeO3 Particles"

_sensors, 2022, doi:10.3390/s22155616_

Round 1

Reviewer 1 Report

The present manuscript is reported VOC sensing properties of the nano-sized p-type semiconducting SmFeO3 particle. The authors have been succeeded in obtaining nano-sized fine SmFeO3 particles by calcining in ozone atmosphere which is effective to oxidize heteronuclear complex a low temperatures, and they realize a selective ethanol detection for the sensor with such particles. However, since they do not compare the sensing property of the present sensor with that of the sensor with conventional SmFeO3 particle obtained by the calcination in oxygen over 900 oC. Therefore, I suggest that the author should add the description about it.

In addition, there are many sentences I cannot understand what the authors intend to write. So, I strongly request that the manuscript should be revised with help of English proofreading service.

Author Response

Thank you for your valuable comments. According to your comment, we have revised our manuscript as follows.

 1. Comment :   However, since they do not compare the sensing property of the present sensor with that of the sensor with conventional SmFeO3 particle obtained by the calcination in oxygen over 900 oC. Therefore, I suggest that the author should add the description about it.

Answer:  The SmFeO3 powder prepared by calciantion at 900oC in air also possesse a good sensing property, but inferior to those calcined in ozone at 600oC in this study. So, we added the comment in line 265-274. 

 2. Comment: In addition, there are many sentences I cannot understand what the authors intend to write. So, I strongly request that the manuscript should be revised with help of English proofreading service.

Answer: Thank you for your comment and we are sorry for our poor English writing. We have revised it with the help of an English proofreading service. 

Reviewer 2 Report

The authors in this paper present sensors based on Nano-sized SmFeO3 particles. The particles were developed by by the pyrolysis of heteronuclear cyano-complex, Sm[Fe(CN)6]ï½¥4H2O at low temperature of 600 degree in ozone. two sensors were prepared using powdered particles with different dimensions resulting in two materials based particles with different specific areas. Sensors were tested with VOCs at different heating temperatures.

1 The authors should add some comments about the reproducibility of coating based particles……how many samples of type A and B have been produced and tested ?

2 The novelty of the paper is not clear , please add comments about it and stress the result related to the novelty concept.

3 in figure 6 the authors should add some comments abouts the signal fluctuation during the curves fall down. Is this phenomenon related to the high value of resistance ?

4 The authors should add some comments or measures about the conductance variations vs the humidity. 

5 Please add a figure and comment about the setup that was used during the performances tes in presence di gasses.

Author Response

Thank you for your valuable comments. According to your comment, we have revised our manuscript as follows.

1 The authors should add some comments about the reproducibility of coating based particles……how many samples of type A and B have been produced and tested ?

 Ansewer: Thank you for your comment. We have repeated for the series of expeirment 3 times and comfired that the classification of sample A and B gave similar fimls of morphology after the sinttering process. This was commented in line 157 to 158.

2 The novelty of the paper is not clear , please add comments about it and stress the result related to the novelty concept.

Answer: Thank you for your comment. We are thinking that a novelty of this work is that down sizing of the SmFeO3 particles by the ozone calcination with a combination of the classificaction process resulted in extraorninary good response and high selectivity to ethanol gas. We added this point in lthe introduction part in line 49-65. 

3 in figure 6 the authors should add some comments abouts the signal fluctuation during the curves fall down. Is this phenomenon related to the high value of resistance ?

Answer: Thank you for your comment. The sensor has a lower resistance in the air than in VOC, but it tends to be more noisy in air. At present, the cause of this is unknown, but the possibility of interference from peripheral devices such as switching timers cannot be ruled out. We added a comment in line 184 to 187. 

4 The authors should add some comments or measures about the conductance variations vs the humidity. 

Thank you for your comment. We measured resisitance of sensors in dry air and 50%RH air. The resistance in dry air was lower than that in 50%RH. We attribbuted this difference to that adsorbed H2O captures hole (h+) of the oxide surface. This was commented in line 169 to 177. Now we are investigating sensor response in dry and humid atmosphere and will report in a next paper.   

5 Please add a figure and comment about the setup that was used during the performances tes in presence di gasses.

Answer: Thank you for your comment. We added a new figure 1 about the setup for the sensor tests, and added a comment in line 95 to 103..

Reviewer 3 Report

Authors synthesized SmFeO3 nanoparticles through low-temperature decomposition. This manuscript presented the VOC sensing properties of this perovskite material. However, the design of the gas sensing experiment is not sufficient. The manuscript also should be carefully checked to avoid some typos. I recommend accepting for publication with minor revision,

1. line 8, should be 'followed by...' 

2. line 33, double 'In'

3. line 52, should move the 'electrophoretic deposition (EPD)' in line 81 to line 52 as EPD showed the first time in the manuscript.

4. Authors need to explain why 280 C and 320 C were selected for gas sensing. 

5. Authors also need to compare the sensing results with other publications.

Author Response

Thank you for your valuable comments and we are sorry for opur gramatical mistakes. Including some minor errors, we have revised our manuscript based on an English prpoofreading service.

1. line 8, should be 'followed by...' 

2. line 33, double 'In'

3. line 52, should move the 'electrophoretic deposition (EPD)' in line 81 to line 52 as EPD showed the first time in the manuscript.

Answer: Thank you for pointing out our mistakes. We corrected them.  

4. Authors need to explain why 280 C and 320 C were selected for gas sensing. 

Answer: Thank you for your comment. We chose the operation temperatures based on sensor resistance which is in the same range of previous studies. This was commented in line 163 to 166.

5. Authors also need to compare the sensing results with other publications.

Answer: Thank you for your comment. We compared the current result with those of other p-type semiconductors such as NiO, Co3O4 and LaFeO3, and SmFeO3 prepred with a different condition. We made new section "3.3.VOC detection using p-type semiconducting oxides" from line 250.

Round 2

Reviewer 2 Report

The authors have answered to the question and they have added some comments in the text. In my opinion now the paper is ready to publish